# Strongly Adhesive and Antimicrobial Peptide-Loaded, Alginate–Catechol-Based Gels for Application against Periimplantitis

Sébastien Baixe [1,2,*], Vincent Ball [1,2,*], Loïc Jierry [3,4], Sarah Cianférani [5], Jean-Marc Strub [5], Youssef Haikel [1,2], Marie-Hélène Metz-Boutigue [1,2] and Olivier Etienne [1,2,*]

1   Faculté de Chirurgie Dentaire, Université de Strasbourg, 8 Rue Sainte Elizabeth, 67000 Strasbourg, France; youssef.haikel@unistra.fr (Y.H.); marie-helene.metz@inserm.fr (M.-H.M.-B.)
2   Institut National de la Santé et de la Recherche Médicale, UMR 1121, 11 Rue Eugène Boeckel, 67085 Strasbourg, France
3   École de Chimie des Polymères et Matériaux, Université de Strasbourg, 25 rue Becquerel, 67087 Strasbourg, France; loic.jierry@ics-cnrs.unistra.fr
4   Centre National de la Recherche Scientifique, Institut Charles Sadron, UPR 22, 23 Rue du Lœss, BP84047, 67034 Strasbourg, France
5   Centre National de la Recherche Scientifique, Laboratory of Bio-Organic Mass Spectrometry, Analytical Sciences Department, Pluridisciplinary Institute Hubert Curien, UMR 7178, Université de Strasbourg, 23 Rue du Loess, 67200 Strasbourg, France; sarah.cianferani@unistra.fr (S.C.); jmstrub@unistra.fr (J.-M.S.)
*   Correspondence: drbaixe@htomail.fr (S.B.); vball@unistra.fr (V.B.); o.etienne@free.fr (O.E.)

**Featured Application: an antibacterial and adhesive hydrogel is developed to inject inside dental implant connections in order to limit bacterial proliferation and periimplant diseases.**

**Abstract:** Background: Periimplantitis is a disease linked to oral virulent bacteria such as *P. gingivalis* that grow in dental implants surrounding tissues and between implants and abutments. Antimicrobial gels previously described to fill these sites lose their effectiveness and resorb over time. Objective: Characterization of biophysical and antimicrobial properties of an original hydrogel, Alginate–Catechol (Alg–Cat), combined to D-Cateslytin (D-CTL). Methods: Gelation kinetics, frequency and strain sweep measurements were performed by rheology. Antibacterial activity of the gels was tested against *P. gingivalis*, and the MIC was determined. Peptides released from the gels were purified by HPLC and characterized by MALDI–TOF mass spectrometry. The behavior of bacteria in contact with the gel was observed using optical and electronic microscopy (SEM and TEM). Results: Gelation was fast and was achieved in 2 min with a storage modulus between 25 and 30 Pa. The gels were stable under strain and showed an adhesive potential reinforced with aging at 18 h (5.4 kPa) under a slow retraction speed (4 J·m$^{-2}$ at 10 μm/s) with a mixed rupture profile (adhesive/cohesive). The MIC of D-CTL inside the Alg–Cat gel against *P. gingivalis* was equal to 470 μg·mL$^{-1}$ after 24 h. Peptides recovered in the supernatant and inside the gel were fragmented, most of them conserving the α-helix active site. No bacteria were visualized at the surface and inside the gel after 24 h. This gel is promising for clinical application for the prevention of periimplantitis.

**Keywords:** hydrogel; AMP; Alginate–Catechol; periimplantitis; *P. gingivalis*; rheology; gelation kinetic; D-cateslytin; chromogranin A; drug release

## 1. Introduction

Nowadays, dental implants are a widely-used option to replace a missing tooth. Basically, assemblies are made of a titanium implant and a crown screwed onto the implant. Due to their biocompatibility, as well as corrosion and mechanical resistence properties, titanium and its derived alloys are considered most suitable for use in implant dentistry [1]. The implant is placed inside the bone and the crown emerges through the gingiva in the oral



cavity. Up to several years after placement of the implant, a marginal bone destruction may occur (periimplantitis) because of bacterial contamination from the teeth to the implant [2]. Indeed, bacterial colonization is considered the main cause of inflammatory events [3,4]. In fact, in some pathological situations, bacteria incriminated in dental plaque may cause an initial inflammatory reaction of the gingiva [5]. When hygiene cannot be well controlled at this step this process can become worse and resorb the periodontal bone surrounding the implant, leading to periimplantitis [6]. Among other etiologies, the hollow spaces inside the implant and the crown can also get contaminated during this process, creating a reservoir of bacteria that can disseminate to living tissues and worsen the periimplantitis [7–10].

To overcome such inflammation processes, gels with intrinsic antimicrobial properties [11,12] have been proposed in medicine and research [13]. These gels could be good candidates to fill the implant/crown connection without obstructing the insertion of components. However, the first oral gels to be proposed incorporated chlorhexidine that only provides a short-term antimicrobial action [14]. Antibiotics have also been used but they may promote resistance or allergic reactions [15,16]. In all cases, the main issues are gel degradation in the oral conditions (due to hydrolysis, pH or temperature variations) and quantitative release of antimicrobial molecules from the gel over time. Among others, gel formulations based on polymers associated to cathecol groups appeared good candidates to use for such an objective.

Different kinds of polymers that are able to undergo gelation were modified with catechol groups for biomedical applications: chitosan [17,18], hyaluronic acid [19], gelatin [20,21], polyethylene glycol [22,23], polyamines [24] and alginate [25]. Their 3D framework is a scaffold that is organized for specific applications such as drug delivery or healing material [25,26]. Gels based on Alginate–Catechol (Alg–Cat) are a promising candidate for this medical use [27]. This combination is bioinspired from holdfast mussel in a wet environment [28] and associates a backbone of alginate grafted with dopamine [29,30]. Synthesis and use of Alg–Cat to build up films was conducted using the layer-by-layer deposition method [30]. In a previous work, we improved the mechanical properties of these gels by adding thiol-modified Pluronic to Alg–Cat in order to induce disulfide-based crosslinks [31]. Here, we chose to evaluate Alg–Cat 1% alone (m/vol) that forms a thinner sticky layer in order to seal the implant/crown connection.

In addition, endogenous antimicrobial peptides (AMPs) are an alternative to antiseptics or antibiotics [32,33]. They are naturally produced by organisms as a first line of defense and belong to the innate immune response [34]. Their amino acid (aa) sequence is generally short with an amphipathic structure, and they are often positively charged, interacting with negatively charged surfaces of bacteria, leading to a penetration of the membrane and the destruction of the microorganisms [35]. Catestatin (CTS) is a 20-aa natural cationic peptide derived from the natural processing of bovine Chromogranin A (CgA$_{344\text{-}364}$) [36] and displays antimicrobial activities against bacteria, yeasts, fungus and parasites [32,37–39]. CTS also contributes to the regulation of the immune system and can modulate a severe inflammatory response [40–46]. The more restricted antimicrobial fragment of catestatin is catestlytin (CTL, CgA$_{344\text{-}3}$), which contains 15 aas and includes the active site of catestatin [36]. The sequence and three-dimensional structure of CTS exhibit homology with penetratin suggesting that both enter the cells by related cell-penetrating mechanisms [41,47]. Furthermore, it was reported that aggregation of CTL beta sheets on negatively charged lipids promote rigid membrane domains [48]. The first introduction of CTL into Alg–Cat gels has shown a bacteriostatic activity, especially on *P. gingivalis* [31]. The combination of Alg–Cat gel with L-CTL has shown biocompatibility with human fibroblasts in a previous study [31].

More recently, to increase the stability of CTL against proteases the D-isomer was prepared and its antimicrobial activities investigated [49,50]. This configuration is not recognized by natural enzymes, especially by *C. albicans* proteases that do not lyse the peptide. D-CTL is also stable in the presence of human saliva [50]. By comparison with L-CTL, the activity was increased against bacteria *(E. coli or S. aureus)* or yeast *(C. albicans)*

and against virulent pathogens found in periimplantitis, especially *P. intermedia and F. nucleatum* [49]. These two bacteria belong to the orange complex of Socransky biofilm organization that represent a virulent group as the red complex with *P. gingivalis*, another highly virulent anaerobic Gram-negative pathogen incriminated in periodontal and periimplant diseases [51]. Moreover, D-CTL emerges as a potent, safe and robust peptide antimicrobial with undetectable susceptibility to resistance and has the remarkable property to potentiate several antimicrobials of reference such as cefotaxime, amoxicillin and methicillin [49].

In the present study, we decided to focus on the mechanical and antibacterial properties of Alg–Cat gels charged with D-CTL by using biophysical, biochemical, microbiological technics and imagery. Finally, the data are discussed in relation with others antibacterial gels.

## 2. Materials and Methods

### 2.1. Chemicals and Gel Preparation

CTL and D-CTL were prepared by ProteoGenix, France. Alg–Cat (structure in Appendix A Scheme A1) was synthesized as previously described [30] with a molar grafting ratio of catechol groups between 10% and 15% (*m/m*) as determined by $^1$H-NMR spectroscopy. The crosslinking occurred starting from dopamine and using EDCl as a crosslinker, as shown in Scheme 1.

**Scheme 1.** Reaction between dopamine and alginic acid to yield Alg–Cat.

Before use, the Alg–Cat powder was put in a closed vessel to limit its oxidation. It was solubilized overnight just before each new experiment in phosphate buffer (50 mM PBS at pH = 7.5) at 1.25% (m/vol at 4 °C). To 4 equivalent volumes of this solution, 1 equivalent volume of NaIO$_4$ (1 mg·mL$^{-1}$) and 0.4 M NaOH were added. Both solutions were vigorously mixed for less than one minute to initiate gelation. For the sake of reproducibility, all the gels were prepared by the same experimentalist. We did not notice variations in the storage modulus larger than 20% from the average for different gel preparations.

The oxidative crosslinking occurs between adjacent dopamine molecules as illustrated in Scheme 2. Crosslinking via Michael addition or Shiff base formation is not possible since alginate does not have nucleophilic sites as with amines or thiols.

Hence, the final concentration in Alg–Cat was 1% (*m/v*). In some experiments, the CTL peptide was added in the initial Alg–Cat solution to reach the expected final load in peptides (from 470 µg·mL$^{-1}$ to 50 µg·mL$^{-1}$). All solutions were prepared in double distilled and demineralized water (MilliQ R04 system, $\rho$ = 18.2 MΩ.cm) and freshly stored at 4 °C for 7 days.

**Scheme 2.** Sodium periodate mediated crosslinking of Alg–Cat chains.

*2.2. Mechanical Evaluation*

For dynamic rheometry measurements, the Alg–Cat and $NaIO_4$ + NaOH solutions were mixed over the lower stainless steel plate of the rheometer to immediately start the measurement of the gelation kinetics. All rheology experiments were performed with a Malvern Kinexus Ultra device, either in the kinetic mode (to measure the gelation kinetics), in the frequency sweep mode (from 10 to 0.01 Hz at 1% of constant strain) or in the amplitude strain mode (from 0.1 to 20% strain at a constant frequency of 1 Hz) to determine the gels' viscoelastic properties. For all rheological measurements, 1.2 mL of gelling material was deposited on the lower plate of the rheometer just after mixing all of the gel components, and the measurements were started 1 min after the end of the mixing process. The gelation kinetics, as well as the amplitude and frequency sweep experiments, were performed with an upper cone that was 4 cm in diameter with an apex angle of 176°. The tacking tests made to investigate the adhesion strength of the Alg–Cat based gels were performed using an upper plate disk that was 2 cm in diameter. The as-deposited gel was aged for a time duration *t* (15 min, 1 h, 4 h or 18 h) with a space of 1 mm between the two stainless steel disks of the rheometer. Before starting the test, the gel was compressed under a force of 1 N for 10 s. The retraction speed was equal to 10, 100 or 1000 $\mu m \cdot s^{-1}$. For the gels that were aged for more than 1 h, water was placed in the surroundings of the rheometer plates to slow down water evaporation from the gel, which could severely modify its adhesion properties.

*2.3. Antimicrobial Properties*

2.3.1. Antimicrobial Activity of Gels Loaded with D-CTL

We evaluated the minimal inhibitory concentration (MIC) of D-CTL against *P. gingivalis* (ATCC® 32277™) (anaerobic Gram-negative bacteria) in Brain Heart Infusion (BHI) (Sigma Aldrich, St. Louis, MO, USA) supplemented with hemin (5 $\mu g \cdot mL^{-1}$, Sigma Aldrich, St. Louis, MO, USA) and menadione (1 $\mu g \cdot mL^{-1}$, Sigma Aldrich, St. Louis, MO, USA). All experiments were conducted under anaerobic conditions in a hermetic box with a sachet generating anaerobic gas (AnaeroGen 2.5 L, Thermo Scientific, Waltham, MA, USA). Before all experiments the gels were rinsed twice with distilled water to induce the desorption of sodium hydroxyde from the gel; this compound was required to speed up the gelation process but by itself is cytotoxic. We did the antimicrobial tests with a volume ratio of 90% bacteria ($OD_{620}$ 0.001)/10% (*m/v*) CTL at different concentrations inside the gel (from 470 $\mu g \cdot mL^{-1}$ to 50 $\mu g \cdot mL^{-1}$). Bacterial test ranges were performed in triplicate in an incubator at 37 °C. At t + 24 h, the optical density ($OD_{620}$) was compared with the control media and 5 $\mu$L were dropped in triplicate on a Petri dish that was put in the incubator at 37 °C. After 24 h, we did a visual control of contaminated dishes: a

bacteriostatic effect of the gel allows a new bacterial growth in contrast with a bactericide activity, which completely kills microorganisms.

In each experiment, Alg–Cat in PBS buffer was mixed with a specific concentration of D-CTL that was freshly diluted in ultra-pure Milli-Q® water. According to previous data relating to a release rate of approximately 10% of AMP outside the Alg–Cat gel [31], we evaluated the optimal concentration of D-CTL to introduce inside the Alg–Cat gel in order to produce the antibacterial hydrogel.

### 2.3.2. Analysis with HPLC and Mass Spectrometry of Remaining CTL Inside the Gel and Its Released Derived Fragments

A measurement of 20 μL from an initial stock solution of L-CTL or D-CTL at 1 mg·mL$^{-1}$ concentration was added to 80 μL of the Alg–Cat initial mixture with 28 μL of gelling solution. A measurement of 180 μL of Milli-Q® water was added to the gel for 24 h at 18 °C. We performed the release experiments in the presence of distilled water and not a physiological buffer in order to simplify the chromatographic analysis, but we are aware that this could slightly modify the gel swelling and hence the peptide release. For the evaluation of the peptide released, 20 μL of the supernatant was directly collected. For the estimation of the remaining CTL inside the gel after 24 h or 7 days, the gel was extracted and crushed with a razor blade in order to increase the surface to volume ratio to ensure quantitative release of the still-incorporated peptide. A measurement of 180 μL of Milli-Q® water was added to the solution of released fragments and after centrifugation for 15 min (13,000 rpm), 20 μL were collected. In all cases, the liquid sample aspired was mixed with 980 μL of a H$_2$0-trifluoroacetic acid (TFA) mixture at 0.1% (vol/vol). The sample was chromatographed with high-performance liquid chromatography (HPLC) and compared with a pure sample of L-CTL/D-CTL from the stock solution. The purity of these peptides was tested by reverse phase (RP) HPLC with a Dionex HPLC system (Ultimate 3000; 13 Sunnyvale, USA) on a Vydac 208 TP C8 column (2.1 × 150 mm$^2$) equipped with a pre-column Vydac 208TP 14 (7.5 × 2.1 mm$^2$) (Vydac, AIT France, Houilles, France). The solvent system consisted of 0.1% (vol/vol) trifluoroacetic acid (TFA) in water (solvent A) and 0.1% (vol/vol) TFA in 70% acetonitrile–water (solvent B) with a flow rate of 0.2 mL/min. The gradient of elution was indicated on chromatograms and each peak detected at $\lambda_{214 \text{ nm}}$ was manually collected. The peptidic material present in each peak was analyzed by mass spectrometry analysis (matrix-assisted laser desorption–time of flight, MALDI–TOF) at the Laboratoire de Spectrométrie de Masse Bio-Organique, UMR7178 (CNRS-UDS, Strasbourg, France). Mass measurements were carried out on an AutoflexTM MALDI–TOF mass spectrometer (Bruker Daltonics GmbH, Bremen, Germany). The matrix solution was prepared from a saturated solution of $\alpha$-cyano-4-hydroxycinnamic acid in water/acetonitrile 50/50 diluted three times in water/acetonitrile/trifluoroacetic acid 50/49.9/0.1 [52].

### 2.3.3. Bacterial Colonization of the Gel

Bacterial colonization of *P. gingivalis* at the surface of and inside the gels was evaluated. Alg–Cat was initially mixed with Milli-Q® water (control), D-CTL (initial concentration 3 mg·mL$^{-1}$) or amoxicillin (1 μg·mL$^{-1}$). *P. gingivalis* (OD 0.001) was layered on Petri dishes and the prepared gel was set on (37 °C, 24 h) before visual observation. All microscopic evaluations were carried out according to a specific protocol. The gels were loaded with L- and D-CTL and directly aspired with a syringe. Contaminated BHI (150 μL) was then poured up to the gel. Assemblies were put in an incubator (37 °C) under anaerobic conditions. At $t$ + 24 h, the supernatant was removed, the gels were recovered and fixed for 48 h (glutaraldehyde 4% adjusted at pH 7.4 + sodium cacodylate 0.05 M), washed three times for 10 min (sodium cacodylate 0.1 M, pH 7.4) and fixed for a second time (sodium cacodylate 0.1 M- osmium tetroxide at 1% m/vol) for 30 min at 4 °C, washed two times for 10 min and then conserved in the wash buffer. For optical and electron transmission microscopy (TEM) (EM 208, FEI, Eidhoven, The Netherlands), the gels were dehydrated, impregnated and embedded with a Spurr resin. For optical microscopy (OM), semi-thin cross sections of 1.25 μm were cut with an ultra microtome,

mounted on a blade and colored with toluidine blue. Ultra-thin slices of 100 nm thickness were applied on metallic grids for TEM. For scanning electron microscopy (SEM) (Hummer JR, Anatech USA, Hayward, CA, USA), the preparation required drying of the gels with a freeze-drying protocol or with the hexamethydisilazane (HMDS) method, putting them in a mixture of absolute ethanol/HMDS (ratio 1:1; vol/vol) for 5 min, then two times into pure HMDS for 5 min. Visualization followed a conventional mode (at 5 kV).

### 3. Results and Discussion

*Gelation Kinetics and Mechanical Properties of the Alg–Cat Based Gels*
Gelation Kinetics

Under the experimental conditions used, the gelation kinetics of Alg–Cat is fast: the storage modulus being higher than the loss modulus after about 2 min of equilibration between the two steel plates of the rheometer (Figure 1a). The G′ value lies between 25 and 35 Pa and it remains constant under deformations up to 50% (Figure 1b), whereas it increases for frequencies above 10 Hz when deformation is constant (Figure 1c), which is characteristic of a viscoelastic material with such a framework [53,54]. We also evaluated the gelling kinetics of D-CTL/Alg–Cat gel in a preliminary assay with increased values of G′ up to 40 Pa (data not shown).

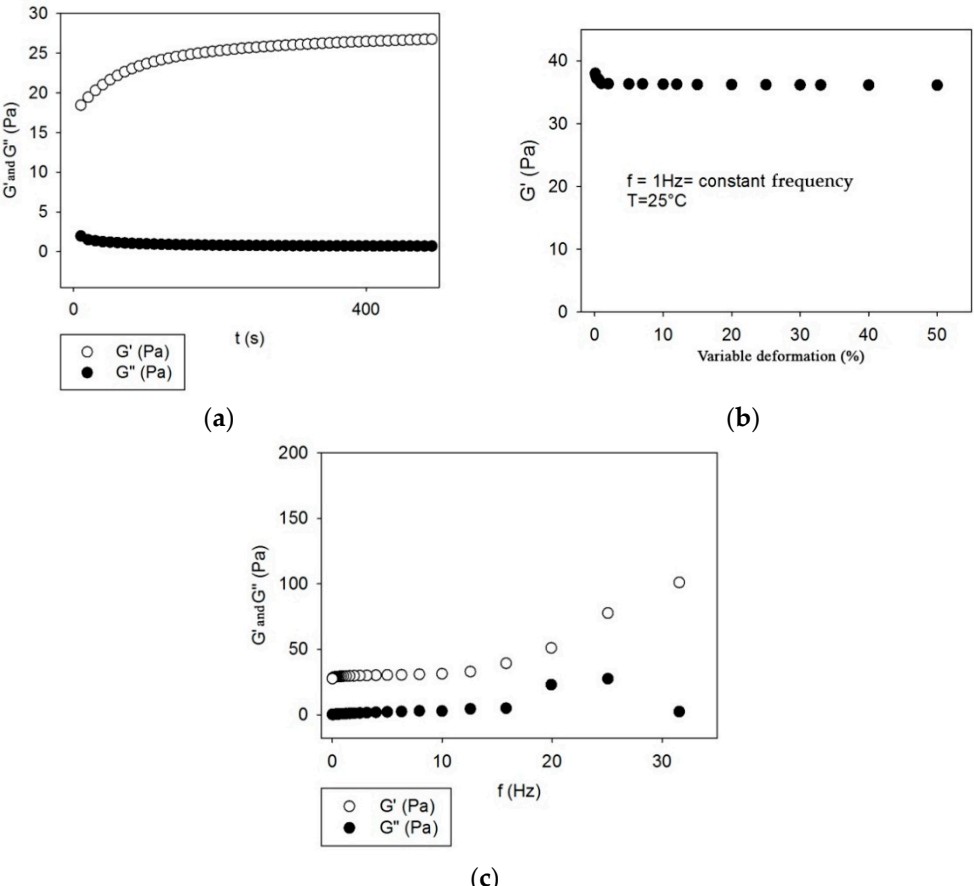

**Figure 1.** Rheological characterizations of Alg–Cat gels. (**a**) Gelation kinetics performed at 1 Hz and a constant strain of 1% displaying the time change in the storage G′ (○) and loss G″ (●) moduli; (**b**) strain sweep experiment performed at a constant frequency of 1 Hz; (**c**) frequency sweep experiment performed at a constant strain of 1% displaying the change in the storage G′ (○) and loss moduli G″ (●).

The maximal adhesion strength of the Alg–Cat hydrogels to stainless steel was obtained after 18 h of aging (Figure 2b) and was equal to 5.4 kPa (1.7 N of the maximal

adhesion force). This value was weaker than the mean shear strength of the PEG-AA-Cat hydrogels on porcine skin, namely (30.4 ± 3.4) kPa [55]. It has to be noted that the four arms of the PEG-AA-Cat hydrogels were modified with catechol groups affording a much higher crosslinking density than our 15% randomly modified Alg–Cat polymer. In addition, Brubaker and Messersmith found that the failure of their connectivity was of cohesive nature whereas it was of mixed nature in our investigation with long gel fibrils breaking or detaching from the substrate during the tests (Figure 2a). As another comparison, Feng et al. measured an adhesion strength of 64.5 kPa in wet conditions when the PEG-Dop (10 kDa in molecular mass) gel was placed between two porcine skin disks at a pull of speed of 17 $\mu m \cdot s^{-1}$ [23]. Note that those experiments were performed at a much higher polymer concentration (10% $m/v$) than in the present study (1% $m/v$). Higher concentrations of Alg–Cat are difficult to prepare because of the viscosity of the formed solution. Adhesive energy is 1 $J \cdot m^{-2}$ for the higher speed of retraction, and up to 4 $J \cdot m^{-2}$ for the lowest speed of retraction, namely 10 $\mu m \cdot s^{-1}$ (Figure 2c). Another strategy to improve adhesion is to add inorganic particles such as calcium phosphate, or to functionalize a titanium surface [56], creating bonds with Alg–Cat. Higher values of G′ for D-CTL/Alg–Cat could be explained by the participation of CTL in the framework of the gel with a highly reactive, negatively charged quinone of catechol with positive charges of CTL, especially on arginine (R). Further investigations need to be conducted in order to confirm these results.

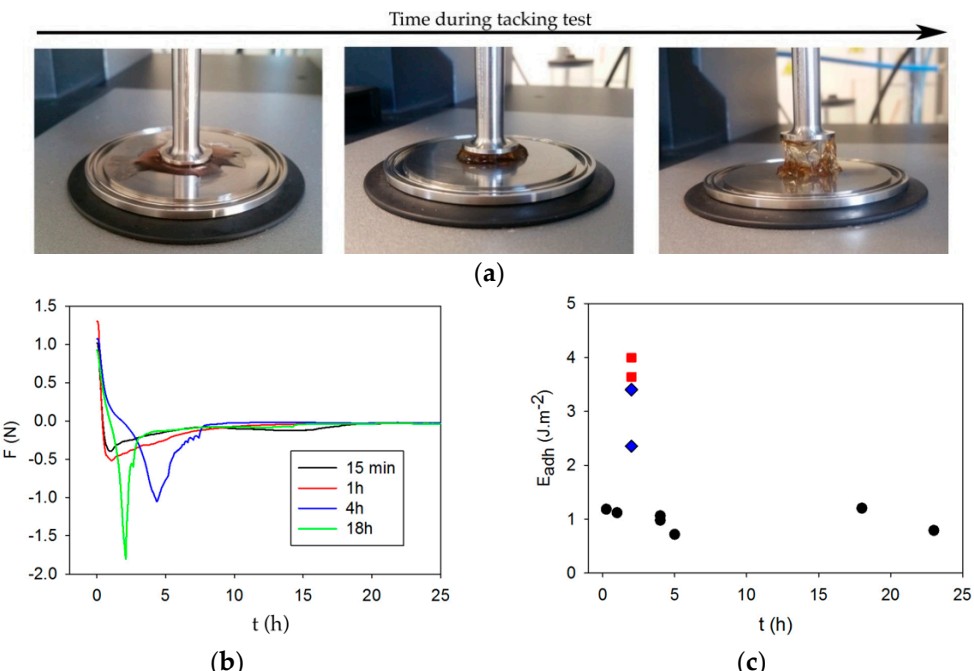

**Figure 2.** Adhesion of Alg–Cat gels onto stainless steel plates. (**a**) Behaviour of the Alg–Cat gel during retraction of the upper plate at a speed of 100 $\mu m \cdot s^{-1}$; (**b**) force versus time curves for the Alg–Cat gels aged for different durations before the tacking tests, performed at a retraction speed of 100 $\mu m \cdot s^{-1}$; (**c**) influence of the aging time and of the retraction speed ((■): 10 $\mu m \cdot s^{-1}$, (◆): 100 $\mu m \cdot s^{-1}$, (●): 1 $mm \cdot s^{-1}$) on the specific adhesion energy of the Alg–Cat gels on steel plates.

## 4. Percentage of *P. gingivalis* Inhibition with D-CTL Inside the Gel

Initial assays show a MIC of D-CTL on *P. gingivalis* of 30 $\mu g \cdot mL^{-1}$ and 200 $\mu g \cdot mL^{-1}$ for L-CTL (data not shown). After OD control, drops sprayed on Petri dishes confirm the result (Figure 3a). This value may be related to that obtained by Zaet and coll. who showed a MIC of 10 $\mu g \cdot mL^{-1}$ for D-CTL (and 149 $\mu g \cdot mL^{-1}$ for L-CTL) against *P. intermedia*, a similar Gram-negative bacteria found in the case of periimplantitis [49]. Hence, D-CTL is encouraging against periimplant pathogens. As shown with these previous results,

combination of AMP and antibiotic inside Alg–Cat gel could decrease the MIC and reinforce the antibacterial activity.

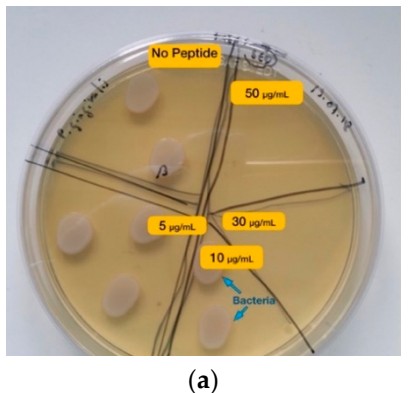

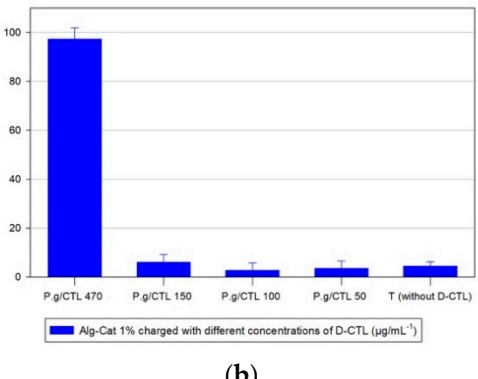

(**a**)                                              (**b**)

**Figure 3.** Determination of the MIC of D-CTL against *P. gingivalis*. (**a**) The MIC of D-CTL in water was estimated to be 30 $\mu$g·mL$^{-1}$ (no recontamination); (**b**) percentage of *P.gingivalis (P.g)* inhibition when bacteria is in contact with gel. The efficient concentration of D-CTL inside gel is 470 $\mu$g·mL$^{-1}$ at more than 24 h (3 mg·mL$^{-1}$ from the stock solution).

Because L-CTL in Alg–Cat gels showed only a bacteriostatic effect against *P. gingivalis* in a previous work [31], we decided to focus on D-CTL for antibacterial evaluation. Bacterial supernatant in contact with the D-CTL (at 470 $\mu$g·mL$^{-1}$) charged gel for 24 h did not grow anymore on the Petri dishes, signing a bactericidal effect of the gel (Figure 3b). Mateescu et al. estimated a release of CTL out of the gel in range of about 10% (*m/m*) of the initially incorporated peptide, namely in this study a concentration of 47 $\mu$g·mL$^{-1}$. It was slightly higher than the measured MIC of 30 $\mu$g·mL$^{-1}$.

Interestingly, it was demonstrated that CTL is not toxic [49] and that it is detected in saliva. Most of the secretory proteins of mammalian salivary glands are mainly synthesized and stored in secretory granules of acinar cells and are extruded by exocytosis into saliva via beta-adrenergic receptor activation. CGA was isolated among these proteins [57]. More recently, it was reported that saliva catestatin could be useful as a biomarker for acute psychological stress in dogs [58].

L-CTL and D-CTL were present in the control group without degradation, mainly in the reduced form (1860 g/mol) (Appendix A Figures A1 and A2a). HPLC analysis shows that peptides released out of the gel, or found inside crushed gel, are not present in the original form (Appendix A Figure A1). Greater peaks analyzed with MALDI–TOF show that CTL is still present, but in a degraded form (Appendix A Figure A2b,c and Table 1). No specific differences between L-CTL and D-CTL fragments were highlighted for all observations. Different molecular weights that were recovered were screened and we tried to affect each of them with a potential sequence of AA derived from the original peptide (web.expasy.org/compute_pi, accessed on 1 October 2021, Table 1). Some of the combinations found were oxidized with one to four oxygen atoms (Figure 4). These oxidation states occur with methionine, phenylalanine and tyrosine residues. CTL-derived peptides were also detected after 7 days (Appendix A Figure A2c). At least five fragments of CTL conserved the LSFR sequence corresponding to the active site of the $\alpha$-helix of CTL, that lightened the remnant antimicrobial of the degraded peptide. In the experimental condition corresponding to 24 h, the different released fragments correspond to 12.5% of the initial D-CTL and 25% of the collected material. Differences of antimicrobial action between L-CTL and D-CTL could be explained by the opposite chiral conformation of the peptides [49]. Cleavages are usually performed on the weakest peptide bond [37] (Figure 4) and we found cleavages of peptidic bonds involving S, G, and A. All these experiments should be carried out by changing pure water that degrades the peptide [59] with human saliva that represents a better physiological condition. Moreover, the resistance of D-CTL to human endoproteases strengthens the interest of this AMP [49].

**Table 1.** Molecular weight as determined by MALDI-TOF mass spectrometry and sequence of the recovered peptides.

| Main Peptides Recovered | Experimental Molecular Weight (*m/z*) | Theorical Molecular Weight (*m/z*) | Theorical Sequences of Peptide (with/without Oxygen) |
|---|---|---|---|
| CTL | 1859.96 | 186.17 | RSM<u>RLSFRA</u>RGYGFR |
| A | 1620.83 | 1620.81 | RSM<u>RLSFRA</u>RGYG + 4 oxydations |
| B | 1563.84 * | 1563.78 | -SM<u>RLSFRA</u>RGYGF + 1 oxydation |
| C | 1448.78 | 1448.62 | -SM<u>RLSFRA</u>RGYG + 3 oxydations |
| D | 1279.64 | 1279.71 | ——RSM<u>RLSFRA</u>R |
| E | 1052.61 | 1052.57 | ——RSM<u>RLSF</u>R |
| F | 989.59 * | 989.49 | —————————<u>FR</u>ARGYGF + 1 oxydation |
| G | 769.44 | 769.41 | —————————<u>FR</u>ARGY |

\* peptide recovered in other samples (data not shown).

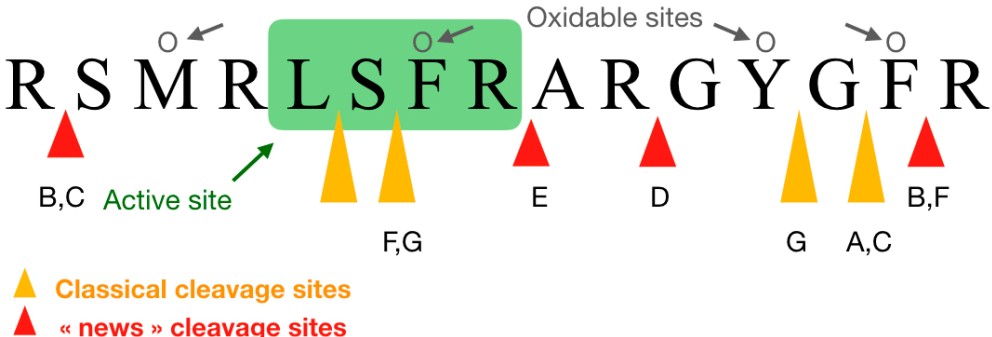

**Figure 4.** Peptides derived from CTL. sequence of L/D-CTL, and specific cleavage sites producing released fragments.

Microscopic views (optical microscopy and SEM) show that the surface of the gel without AMP is completely colonized by *P. gingivalis* at 24 h (Figures 5a and 6c,d), and TEM confirms these results (Appendix A Figure A3). Bacteria grow on the whole surface of the gel but do not grow inside the bulk, this was certainly because of the strengthened framework. Adjunction of D-CTL inside the gel impedes bacterial colonization (Figures 5b and 6e,f) after 24 h. Hence, it is a candidate for early healing inside implant cavities but also for other medical applications. A longer time of in vitro evaluation is needed to determinate the maximum time of antibacterial action at the surface and if bacteria grow inside the gel. In this way, we could see if a part of CTL, which certainly participates in the structure of the gel, is released during aging in a wet environment.

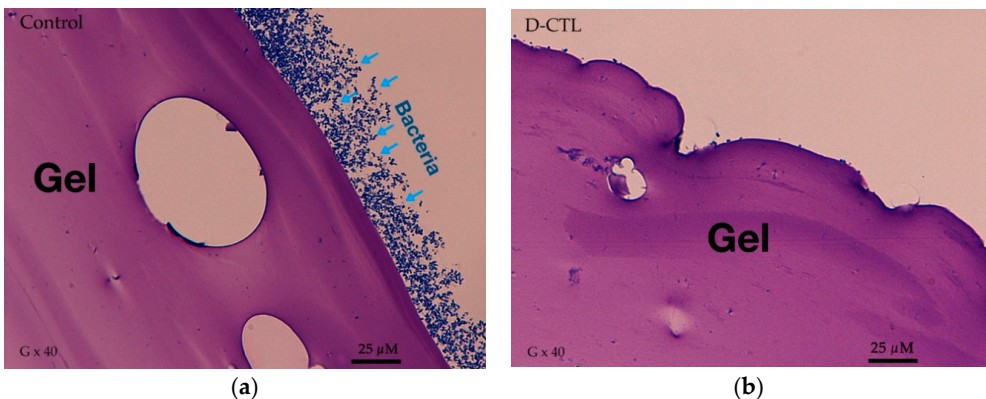

(**a**)  (**b**)

**Figure 5.** Alg–Cat 1% without/with antimicrobial agent and *P. gingivalis* (optical microscope view). (**a**) Gel without antibacterial agent is recovered by bacteria at the surface at 24 h (**b**) when D-CTL is added inside the gel, no bacteria are observed, showing an antibacterial effect.

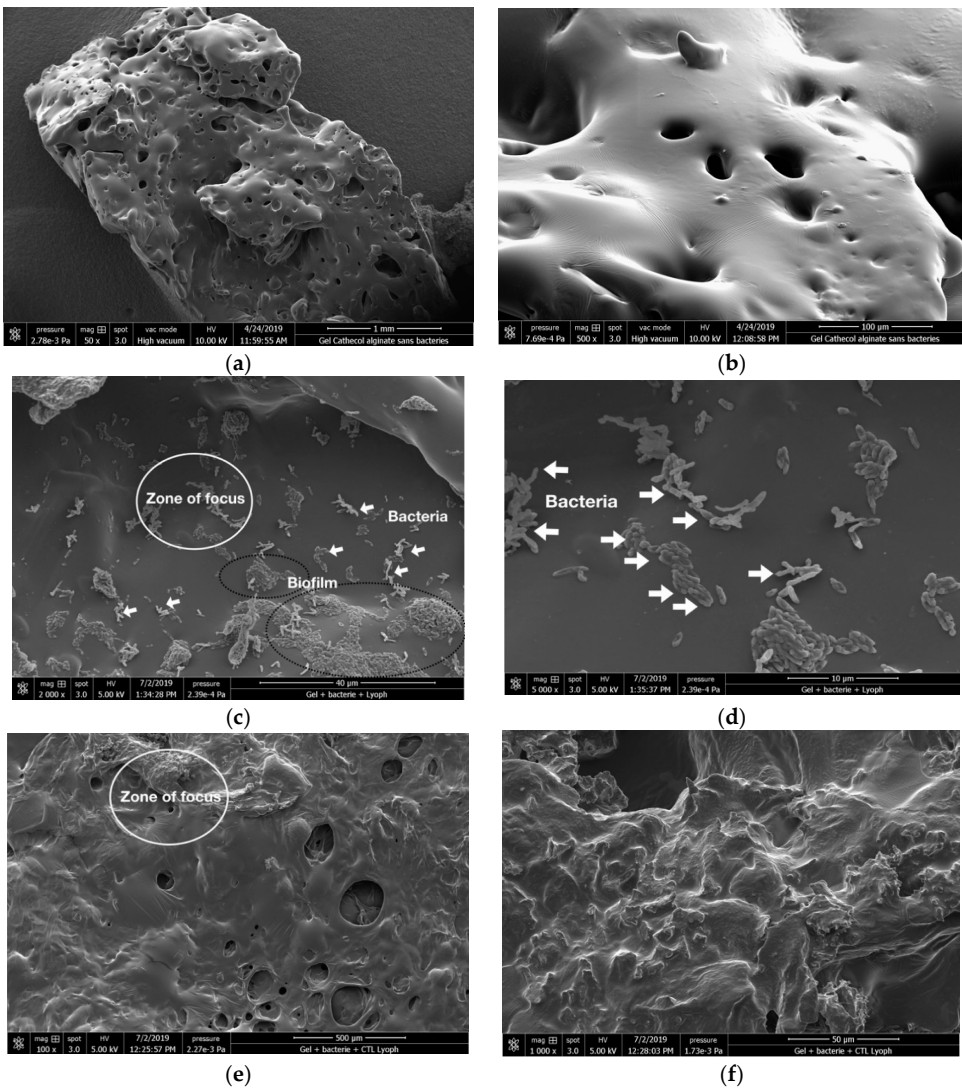

(**a**)  (**b**)

(**c**)  (**d**)

(**e**)  (**f**)

**Figure 6.** Alg–Cat 1% with/without antimicrobial agent and *P. gingivalis* (MEB view). (**a**,**b**) Gel presents a porous structure that could be layered by biofilm; (**c**,**d**) *P. gingivalis* colonize the surface and pores when an antibacterial agent is not present; (**e**,**f**) when charged with D-CTL, no bacteria were found at 24 h.

## 5. Conclusions

The gelation time of the Alg–Cat was found to be lower than 2 min, which is highly compatible with a clinical use. Its viscoelasticity and adhesive properties, enable its insertion inside implant connections without interfering with an accurate placement of the prosthetic components. D-CTL may reinforce the framework of Alg–Cat gel and provide a bactericidal activity at 24 h.

The different released fragments conserve the active site of D-CTL and display an interesting antimicrobial activity. For a short time, bacteria do not grow at the surface and inside the Alg–Cat gel, which constitutes a physical–chemical barrier against *P. gingivalis*. Other bacteria as well as a longer-term evaluation should be investigated to confirm the Alg–Cat/D-CTL hydrogel potential.

**Author Contributions:** Conceptualization, Y.H., O.E., M.-H.M.-B. and V.B.; methodology, M.-H.M.-B., V.B. and J.-M.S.; validation, O.E., M.-H.M.-B., V.B. and J.-M.S.; investigation, S.B., V.B., L.J. and S.C.; writing—original draft preparation, S.B., V.B., M.-H.M.-B. and O.E.; writing—review and editing, S.B., V.B., M.-H.M.-B. and O.E.; supervision, O.E., V.B., M.-H.M.-B. and Y.H. All authors have read and agreed to the published version of the manuscript.

**Funding:** This research received funding from the MICA project.

**Data Availability Statement:** Data contained within the article or supplementary material are available on request from the corresponding author.

**Acknowledgments:** We are indebted to Éric Mathieu and Naji Kharouf for their contribution in the SEM experimentations, to Abdurraouf Zaet and Sophie Hellé for their help in microbiology and to Cosette Betscha for HPLC analysis.

**Conflicts of Interest:** The authors declare no conflict of interest.

## Appendix A

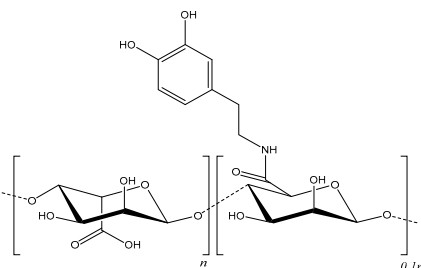

**Scheme A1.** Structure of the Alg–Cat polymer used in this study.

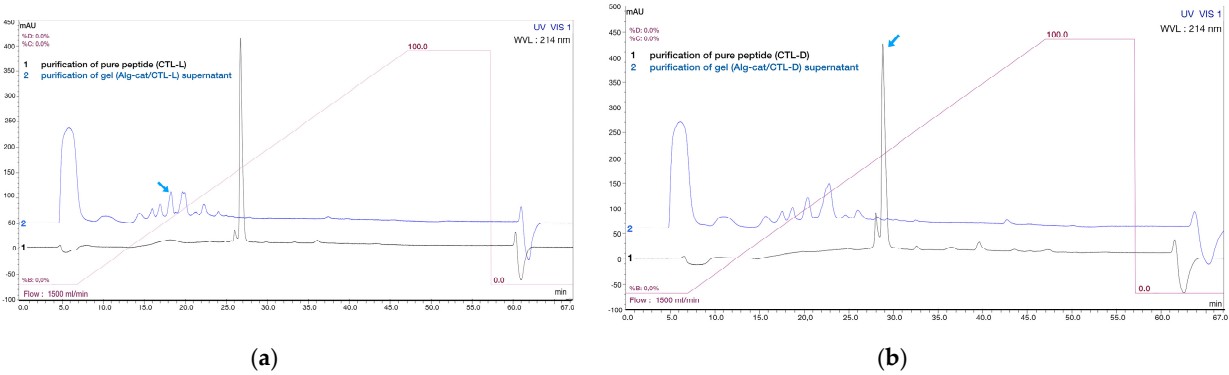

<div align="center">(a)        (b)</div>

**Figure A1.** HPLC analysis of CTL (at + 24 h). Black curves (**1**) show a main peak corresponding to the reduced and active form of L-CTL (**a**) and D-CTL (**b**). Profiles of curves are similar between L and D-CTL. Blue curves (**2**) display a main peak (salts), followed by several secondaries (seven) peaks that correspond to peptides in the supernatant of the gel after 24 h (and it is similar after 7 days). The complete characterization of the fragments indicated by blue arrows (↓) is obtained with MALDI–TOF analysis (Appendix A Figure A2).

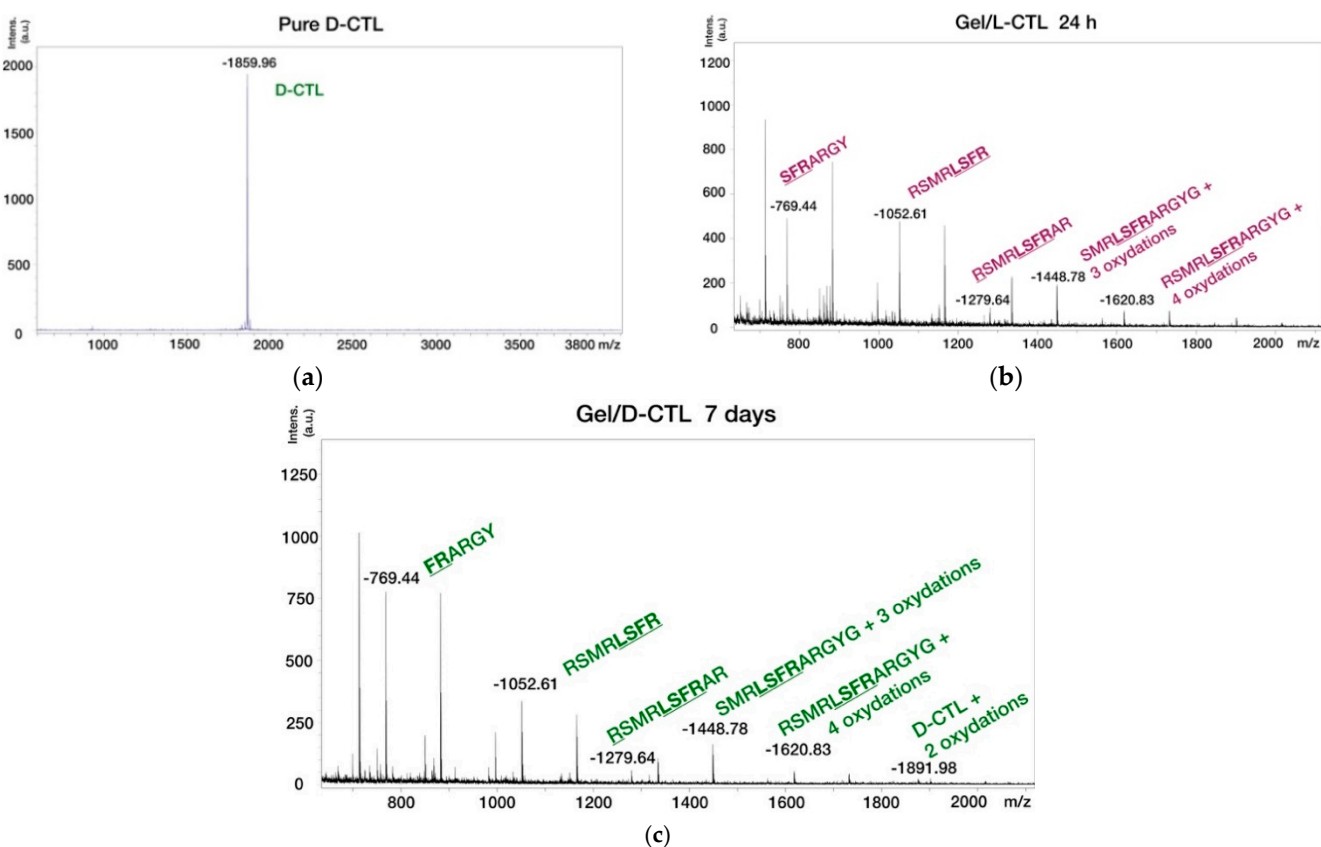

**Figure A2.** MALDI–TOF analysis. (**a**) Complete sequence of CTL is found in all control samples. (**b,c**) Fragments of degraded peptide are found in the supernatant of the gel at 24 h (**b**) and 7 days (**c**), for L-CTL and D-CTL.

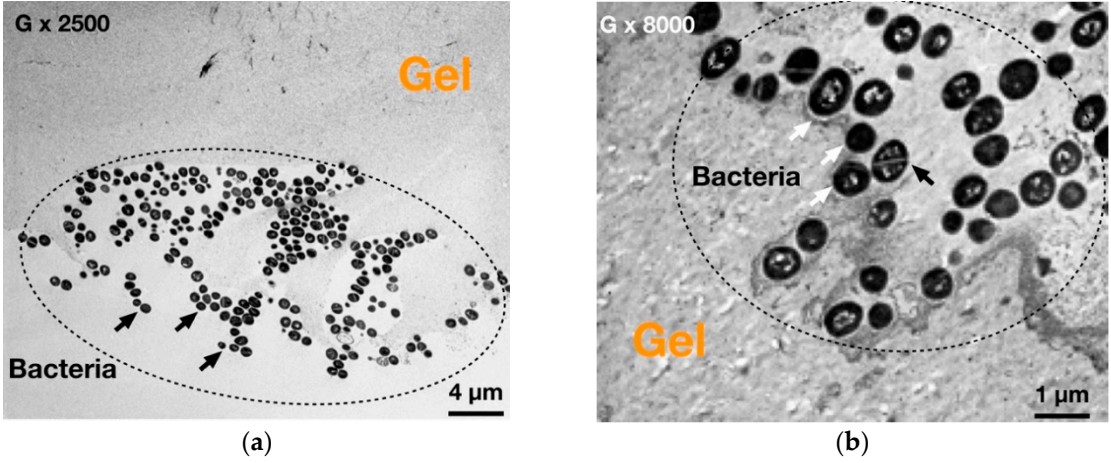

**Figure A3.** TEM views of Alg–Cat gel surface contaminated with *P. gingivalis*. (**a,b**) Bacteria colonize the surface of the gel and porous structure, but do not grow inside gel.

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
