# Peer review of "Strongly Adhesive and Antimicrobial Peptide-Loaded, Alginate–Catechol-Based Gels for Application against Periimplantitis"

_applsci, doi:10.3390/app112110050_

Round 1
Reviewer 1 Report
The work entitled “Strongly adhesive and antimicrobial peptide loaded alginate-catechol based gels for application against periimplantitis” by Baixe et al. aimed at characterizing the biophysical and antimicrobial properties of a hydrogel made of alginate-catechol loaded with D-Cateslytin. As the authors stated, this is the first report of its kind, so the innovative nature of this research is clear. The authors did an excellent job of introducing the subject and detailing the methodology employed. The discussion and the presentation of the results is well done as well. However, it would be great if a little more literature was used to confront the results. This is the only lacunae of this otherwise good research. I would recommend the authors to deepen this part of their discussion prior to publication.
Author Response
The work entitled “Strongly adhesive and antimicrobial peptide loaded alginate-catechol based gels for application against periimplantitis” by Baixe et al. aimed at characterizing the biophysical and antimicrobial properties of a hydrogel made of alginate-catechol loaded with D-Cateslytin. As the authors stated, this is the first report of its kind, so the innovative nature of this research is clear. The authors did an excellent job of introducing the subject and detailing the methodology employed. The discussion and the presentation of the results is well done as well. However, it would be great if a little more literature was used to confront the results. This is the only lacunae of this otherwise good research. I would recommend the authors to deepen this part of their discussion prior to publication.
Our reply: We acknowledge reviewer 1 for his/her positive and constructive comments about our article.
Our replies concerning the comments to this reviewer appear in blue in the revised version of the manuscript.
We totally agree with the comments and we add 3 references at the level of the Introduction (the references appear in the reference list under the number 18, 21, 24). All the other reference numbers are updated in the main body of the text as well as in the reference list.

Reviewer 2 Report
Comments on Baixe et al:
The aim of this study is to characterize the biophysical and antimicrobial properties of an original hydrogel Alginate-Catechol combined to D-Cateslytin, using biophysical, biochemical and microbiological techniques.
Even if the manuscript provides an organic overview, with a densely organized structure and based on well-synthetized data, there are aspects to be mentioned, to make the article fully readable. For these reasons, the manuscript requires minor changes.
Please find below an enumerated list of comments on my review of the manuscript:
LINE 41: Due to their biocompatibility, as well as corrosion and mechanical resistence properties, Titanium and its derived alloys are considered most suitable to be used in implant dentistry (see, for reference: Microbiological and SEM-EDS Evaluation of Titanium Surfaces Exposed to Periodontal Gel: In Vitro Study – 2019; Carbon Coatings Deposited on Prosthodontic Ni-Cr Alloy – 2021).
LINE 45: Bacterial colonization is considered the main cause of inflammatory events, as highlighted by recent and different studies (see, for reference: Commercial oral hygiene products and implant collar surfaces: Scanning electron microscopy observations – 2020; Poor oral Hygiene may be the Sole Cause of Oral Cancer – 2012).
MATERIAL AND METHODS:
As regards this section, the methodology design was rigorous and appropriately implemented within the study.
RESULTS AND DISCUSSION:
Also this section is well organized and densely presented, based on well-synthetized data.
In conclusion, this manuscript is densely presented and well organized, based on well-synthetized data. Overall, the contents are rich and the authors also give their deep insight for some works. The authors were lucid in their style of writing, making it easy to read and understand the message, portrayed in the manuscript. The article is very informative, even if requires few changes (as mentioned). It would be better if they include some important and recent contributions in the manuscript. I would accept the manuscript, if the comments are addressed properly.
Author Response
Our reply: We acknowledge this reviewer for her/his constructive comments. We add some text between lines 45 and 51 to reply to the comment concerning the importance of titanium based implants and bacteria. In addition we add three additional references (ref 1, 3 and 4in the revised version) as requested. Our comments concerning this reviewer appear in green in the revised manuscript.

Reviewer 3 Report
The publication is written in a proper way. The introduction is comprehensive, the literature references are current and sufficient. Imperfections appear in the description of the methodology and this should be corrected before the paper is published.
Chapter 2.1. Chemicals and gel preparation” The publication lacks a description of the synthesis of the alginate derivative, there is only a reference in the literature. Similarly, the reactions responsible for gel crosslinking are also not sufficiently described. There should be schemes of the reactions occurring.
Line 117 sentence „Both solutions were vigorously mixed to initiate gelation.” How the mixtures were mixed, for how long, and with what force. These are details that will affect the properties of the gels.
The material consists of a sodium base that has not been removed from the finished gel. This is a component that will be toxic. The authors should comment on this somehow and include it in the controls of the biological experiments.
Line 166 the release experiment was conducted in water, why not in an isotonic solution closer to physiological conditions, e.g. PBS.
Linia 168 „the gel was extracted and crushed with a razor blade” How the gel is fragmented will affect the release. If the authors choose to do this scenario then please describe it in more detail . Does the gel need to be fragmented for such an experiment?
Authors need to address these comments before publication.
Author Response
We acknowledge reviewer 3 for his/her constructive comments about our article. Our replies concerning his/her comments appear in red in the revised manuscript (attached file)
*Point 1: Chapter 2.1. Chemicals and gel preparation” The publication lacks a description of the synthesis of the alginate derivative, there is only a reference in the literature. Similarly, the reactions responsible for gel crosslinking are also not sufficiently described. There should be schemes of the reactions occurring.
Our reply: we totally agree with the reviewer and we add two schemes in the revised article. Scheme 1 describes the synthesis of the modified alginate whereas Scheme 2 illustrates the chemical reaction occurring during the sodium periodate crosslinking.
Some text is also added on lines 125-127 of the revised manuscript: "The oxidative crosslinking occurs between adjacent dopamine molecules as illustrated in Scheme 2. Crosslinking via Michael addition or shiff base formation is not possible since alginate doesn't have nucleophilic sites like amines or thiols."
*Point 2: Line 117 sentence „Both solutions were vigorously mixed to initiate gelation.” How the mixtures were mixed, for how long, and with what force. These are details that will affect the properties of the gels.
Our reply: We perfectly agree with the reviewer and we clarify this point in the revised manuscript : “Both solutions were vigorously mixed for less than one minute to initiate gelation.” (line 121). Indeed, under these conditions, the gelation is extremely fast.
The applied force was not controlled, but all the experiments were performed by the same author of the article to ensure reproducibility. We did not notice variations in the storage modulus higher than 20 % from one experiment to the other. A sentence is added on lines 122-124 in the revised manuscript: “For the sake of reproducibility, all the gels were prepared by the same experimentalist. We did not notice variations in the storage modulus larger than 20 % from the average for different gel preparations.”
*Point 3: The material consists of a sodium base that has not been removed from the finished gel. This is a component that will be toxic. The authors should comment on this somehow and include it in the controls of the biological experiments.
Our reply: We agree with the reviewer, but sodium hydroxide was required to improve the sodium periodate induced crosslinking. The gel was washed with water before the experiments in contact with the bacteria. We add a sentence on line of the revised manuscript, on line 162-164: "Before all experiments the gels were rinsed two times with distilled water to induce the desorption of sodium hydroxyde from the gel, this compound being required to speed up the gelation process but being by itself cytotoxic."
*Point 4: Line 166 the release experiment was conducted in water, why not in an isotonic solution closer to physiological conditions, e.g. PBS.
Our reply: basically we agree with the reviewer. But we used water to simplify the chromatographic analysis of the released peptides. One cannot exclude that the release profile could have been slightly different in the presence of a buffer like PBS. We add a sentence on line of the revised manuscript, lines 180-182: “We performed the release experiments in the presence of distilled water and not a physiological buffer in order to simplify the chromatographic analysis but we are aware that this could slightly modify the gel swelling and hence the peptide release
*Point 5: Line 168 „the gel was extracted and crushed with a razor blade” How the gel is fragmented will affect the release. If the authors choose to do this scenario then please describe it in more detail . Does the gel need to be fragmented for such an experiment?
Our reply: We agree with the reviewer and we complete the sentence on lines 185-186 (in the revised version) by adding : “in order to increase the surface to volume ratio to ensure quantitative release of the still incorporated peptide.”

Round 2
Reviewer 3 Report
The authors have satisfactorily addressed the reviewer's comments. The work has benefited and can be published in this form.